# Diacerein provokes apoptosis, improves redox balance, and downregulates PCNA and TNF-α in a rat model of testosterone-induced benign prostatic hyperplasia: A new non-invasive approach

Rabab Ahmed Rasheed [1]*, A. S. Sadek[2,3], R. T. Khattab[2], Fatma Alzahraa A. Elkhamisy[4], Heba Abdelrazak Abdelfattah[5], Mohamed M. A. Elshaer[6,7], Saeedah Musaed Almutairi[8], Dina S. Hussein[9], Azza Saleh Embaby[10], Mai A. M. Almoatasem[10]

1 Department of Medical Histology and Cell Biology, Faculty of Medicine, King Salman International University, South Sinai, Egypt, 2 Department of Anatomy and Embryology, Faculty of Medicine, Ain Shams University, Cairo, Egypt, 3 Department of Anatomy and Embryology, Faculty of Medicine, King Salman International University, South Sinai, Egypt, 4 Department of Pathology, Faculty of Medicine, Helwan University, Cairo, Egypt, 5 Department of Histology and Cell Biology, Faculty of Medicine, Helwan University, Cairo, Egypt, 6 Department of Clinical Pharmacology, Faculty of Medicine, Ain Shams University, Cairo, Egypt, 7 Department of Clinical Pharmacology, Faculty of Medicine, King Salman International University, South Sinai, Egypt, 8 Department of Botany and Microbiology, College of Science, King Saud University, Riyadh, Saudi Arabia, 9 Department of Chemistry, College of Sciences and Health, Cleveland State University, Cleveland, United States of America, 10 Department of Medical Histology & Cell Biology, Faculty of Medicine, Beni-Suef University, Beni-Suef, Egypt

* rabab.rasheed@ksiu.edu.eg

**Data Availability Statement:** All relevant data are within the paper.

## Abstract

One of the most prevalent chronic conditions affecting older men is benign prostatic hyperplasia (BPH), causing severe annoyance and embarrassment to patients. The pathogenesis of BPH has been connected to epithelial proliferation, inflammation, deranged redox balance, and apoptosis. Diacerein (DIA), the anthraquinone derivative, is a non-steroidal anti-inflammatory drug. This study intended to investigate the ameliorative effect of DIA on the prostatic histology in testosterone-induced BPH in rats. BPH was experimentally induced by daily subcutaneous injection of testosterone propionate for four weeks. The treated group received DIA daily for a further two weeks after induction of BPH. Rats' body and prostate weights, serum-free testosterone, dihydrotestosterone, and PSA were evaluated. Prostatic tissue was processed for measuring redox balance and histopathological examination. The BPH group had increased body and prostate weights, serum testosterone, dihydrotestosterone, PSA, and oxidative stress. Histologically, there were marked acinar epithelial and stromal hyperplasia, inflammatory infiltrates, and increased collagen deposition. An immunohistochemical study showed an increase in the inflammatory TNF-α and the proliferative PCNA markers. Treatment with DIA markedly decreased the prostate weight and plasma hormones, improved tissue redox balance, repaired the histological changes, and increased the proapoptotic caspase 3 expression besides the substantial reduction in TNF-α and PCNA expression. In conclusion, our study underscored DIA's potential to alleviate

**Funding:** The authors extend their appreciation to the Researchers supporting project number (RSP2023R470) King Saud University, Riyadh, Saudi Arabia.

**Competing interests:** The authors have declared that no competing interests exist.

**Abbreviations:** BPH, Benign prostatic hyperplasia; CMC-Na, Sodium carboxymethylcellulose 0.5%; DHT, Dihydrotestosterone; DIA, Diacerein; GSH, Glutathione enzyme; H&E, Hematoxylin & eosin; MDA, Malondialdehyde enzyme; PCNA, Proliferating cell nuclear antigen; PSA, Prostatic specific antigen; ROS, Reactive oxygen species; SOD, Superoxide dismutase; TNF-α, Tumor necrosis factor-alpha; TP, Testosterone propionate.

the prostatic hyperplastic and inflammatory changes in BPH through its antioxidant, anti-inflammatory, antiproliferative, and apoptosis-inducing effects, rendering it an effective, innovative treatment for BPH.

## 1. Introduction

One of the most prevalent chronic conditions affecting older men is benign prostatic hyperplasia (BPH), whereas incidence is markedly increased among men above 50 [1]. Although BPH is considered a benign process, the fact that it might progress to a malignant condition, such as prostate carcinoma, attracts great concern [2]. Also, lower urinary tract disorders like frequent urination, urinary infections, and retention are usually linked to BPH, causing severe annoyance and embarrassment to patients, thus leading to decreased life quality [3]. On the histologic level, BPH is defined as prostatic tissue overgrowth involving epithelial and stromal cells [4].

Growth factors are often generated by the prostate's stromal cells, which use autocrine and paracrine pathways to keep prostate cells in a condition of homeostasis. The root cause of BPH development is a change in the equilibrium of cellular homeostasis. The androgen receptors become more active, increasing the proliferation-promoting growth factors [5]. BPH's exact etiology has not yet been fully revealed, but many theories have correlated its pathogenesis to cell proliferation, ROS accumulation, inflammation, apoptosis, and hormonal imbalance [5–7].

Dihydrotestosterone (DHT), the byproduct of the conversion of testicular testosterone by 5-alpha-reductase-2 in the prostatic stromal cells, has not only direct proliferative effects on these prostatic cells but also has endocrine effects in the bloodstream, which can affect cellular proliferation and apoptosis [8]. The levels of DHT, which strongly correlate with BPH, can be decreased by the 5-red-2 inhibitors [9].

The two main factors that control cell development are apoptosis and inflammation. BPH is associated with apoptosis dysregulation. Increased levels of Dickkopf-related protein 3 in BPH pathophysiological circumstances result in a decrease in key apoptotic proteins such B cell lymphoma (Bcl-2) associated X protein (Bax). Additionally, it prevents caspase activation and influences cell death via modifying nuclear factor-κB [10, 11].

Chronic inflammation has been proven to cause excessive prostate cell growth in BPH. Under the influence of BPH, immune infiltrate cells become chronically activated and produce cytokines, including TNF-α, IL-1, and IL-6. These cytokines support the expansion of epithelial cells and enlargement of the prostate [12, 13]. Certain inflammatory pathways can be activated by oxidative stress via activating transcription factors like NF-κB, which in turn regulate the gene expression of proinflammatory mediators, including TNF-α [14].

Moreover, by upsetting the prostate's normal equilibrium between cell growth and cell death, prostatic inflammation can cause stromal and glandular hyperplasia. Similarly, testosterone is known to cause alterations in the prostate gland's cycline-D1, PCNA, Bax, and Bcl2 mRNA levels that are pro-proliferative and anti-apoptotic [15].

Nowadays, 5α-red 2 inhibitors and α-blockers are the most recommended drugs for treating BPH [16]. The α-blockers relax the prostatic smooth muscles, thus relieving BPH symptoms [17]. However, both drugs showed various adverse effects, including erectile dysfunction, cardiovascular hazards, and dizziness [18]. Therefore, searching for and using "safer" and effective alternatives against BPH is necessary.

DIA, an anthraquinone derivative, is an anti-inflammatory, analgesic, and antipyretic drug used mainly in the treatment of osteoarthritis [19]. DIA is metabolically converted into rhein, which has a wide range of significant properties, including anticancer, antioxidant, and anti-inflammatory effects [20]. The anti-inflammatory properties of rhein are mainly due to preventing the synthesis of several cytokines, including TNF-α, IL-1, and IL-6. It has also been observed to knock down NF-κB pathway signaling [21]. Moreover, DIA was able to normalize serum testosterone, correcting testicular histological alterations and lowering oxidative stress parameters produced by ischemia-reperfusion testicular damage [22]. Additionally, Oliveira and colleagues reported rhein's antimicrobial, chondroprotective, and antidiabetic characteristics [23].

To the best of our knowledge, this work is one of few that explored the putative ameliorative effect of DIA on testosterone-induced-BPH in rats on the biochemical, histologic, immunohistochemical, and ultrastructural levels, which may be linked to its antioxidant, anti-inflammatory, antiproliferative, and apoptosis-inducing effects.

## 2. Material and methods

### 2.1. Drugs and materials

Testosterone propionate (TP) (oily solution in ampoules 250 mg/mL) was provided from El-Nile Company for pharmaceuticals and Chemical Industries, Egypt. Diacerein (DIA) (capsules 50 mg) was provided from EVA Pharma, Egypt. Sodium carboxymethylcellulose 0.5% (CMC-Na) solution obtained from Biochemistry Department, School of Medicine, Cairo University, Egypt. Pure olive oil purchased from Seoudi Market, Giza, Egypt.

### 2.2. Animals housing and ethical declaration

The current study's animal model consisted of Sprague Dawley rats (200–222 g, 5-6-months-old) purchased from the National Research Centre (NRC), Cairo. According to the NIH Guidelines for the Care and Use of Laboratory Animals, the study was conducted at the Animal House of the Faculty of Pharmacy, Beni-Suef University. Rats were allowed to acclimate for one week prior to the experiment in a pathogen-free, naturally ventilated environment. Rats were sheltered in polycarbonate cages containing six animals each, in a temperature ranging between 22:24°C. In a light-controlled room (12:12 h light-dark cycle), rats had unlimited access to chow and tap water. The Institutional Animal Care and Use Committee, Beni-Suef University, Egypt (BSU-IACUC), accepted the study protocol (Code: 022–367).

### 2.3. Experimental design

For six weeks after the acclimation period, forty rats were indiscriminately assigned to four groups (n = 10) as follows: vehicle control: received olive oil by s.c injection daily for four weeks, followed by 0.5% CMC-Na solution via oral gavage for a further two weeks; DIA group: received olive oil by s.c injection daily for four weeks, followed by DIA dissolved in 0.5% CMC-Na solution (50 mg/kg daily) via oral gavage [24] for further two weeks; BPH group: received TP dissolved in olive oil (5 mg/kg/day, s.c) for four weeks for the induction of BPH [25], followed by 0.5% CMC-Na solution via oral gavage for another two weeks; and BPH +DIA group: received TP for four weeks followed by DIA for further two weeks after the induction of BPH at the previously specified doses. Generally, all medications were administered daily, and the animals' body weights were assessed and recorded once weekly at a fixed time. The present study's design and timeline are illustrated in Fig 1.

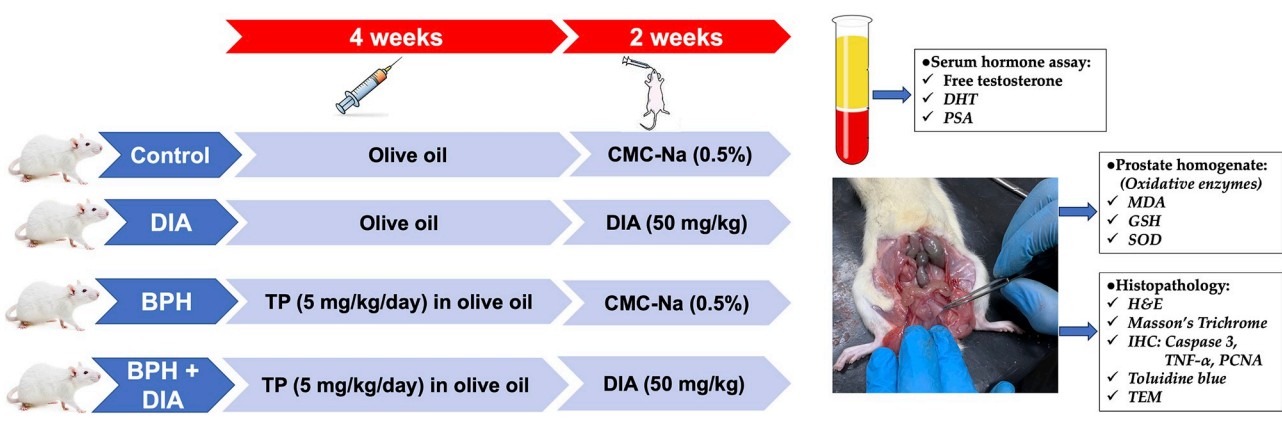

**Fig 1. The design and the timeline of the study.**

At the end of the experiment (6 weeks), the rats' body weights were recorded. After an overnight fast, blood samples were obtained from the tail veins and centrifuged (3000 rpm, 5 min) to separate the serum for hormonal assay. Rats were then euthanized by ketamine (300mg/kg) and i.p injection of xylazine (30mg/kg) [26], followed by cervical dislocation. A laparotomy incision was done, and the prostates were extracted from all the rats and weighed. Specimens from the ventral lobe of each prostate were excised and immediately processed for the histopathological assessment; the remaining prostatic tissue was homogenized in phosphate-buffered solution 0.1 M (pH 7.4) using a Teflon homogenizer and centrifuged. The supernatant was collected in Eppendorf containers, frozen in liquid nitrogen, and kept at −80˚C for further biochemical lab studies.

## 2.4. Measurement of body weight, prostate weight, and prostate index

The rats' body weight was recorded prior to the beginning of the experiment and continued once weekly using a digital scale. The prostate weight was assessed after sacrificing the animals. Prostate index of each rat was estimated according to the following formula: (weight of the prostate/total body weight) × 1000 [15].

## 2.5. Evaluation of serum testosterone, dihydrotestosterone (DHT), and Prostatic Specific Antigen (PSA)

Free testosterone, DHT, and PSA levels were assessed in serum according to the manufacturer's guidelines of the corresponding ELIZA kits (MyBioSource Cat# MBS740121, SAN Diego, USA), (SunLong Biotech Cat# SL0236Ra, China), (SunLong Biotech Cat# SL0605Ra, China), respectively.

## 2.6. Evaluation of oxidative stress in prostatic tissue homogenate

The tissue content of Malondialdehyde (MDA), Glutathione (GSH), and Superoxide Dismutase (SOD) activity was biochemically estimated in prostatic tissue homogenate by colorimetric method using Bio-diagnostic kits; Cat# MD 25 29, Cat# GR 25 11, and Cat# SD 25 21, Giza, Egypt respectively, according to the manufacturer's guidelines.

## 2.7. Histopathological examination

**2.7.1. Light microscopic examination.** Specimens from the ventral lobe of the prostate of each rat from all study groups were immediately fixed in a neutral-buffered formol solution 10%, dried in graded ethanol, and then immersed in molten wax to obtain paraffin blocks. Thin sections (4 μm) were cut and dyed with hematoxylin and eosin (H&E) and Masson's trichrome stains as per [27].

**2.7.2. Immunohistochemical staining for caspase 3, Tumor Necrosis Factor-Alpha (TNF-α), and Proliferating Cell Nuclear Antigen (PCNA).** Prostatic paraffin sections were processed for the immunohistochemical staining based on the peroxidase-avidin-biotin method [27]. Sections were deparaffinized, rehydrated, and then blocked for peroxidase enzyme using $H_2O_2$ (3%) for ten minutes. For another ten minutes, the slides were washed using phosphate-buffered solution (pH 7.4), then immersed in 1% bovine serum at 37˚C, aiming to diminish the non-specific reactions. After heating in citrate buffer for antigen retrieval, the following antibodies were utilized as primary antibodies: anti-caspase 3 as a proapoptotic marker (mouse, monoclonal IgG, diluted 1:100, Cat# MA1-16843, Thermo Fisher Scientific, CA, USA), anti-TNF-α as an inflammatory marker (rabbit, polyclonal IgG, diluted 1:50, Cat# PA1-40281, Thermo Fisher Scientific, CA, USA), and anti-PCNA as a proliferative marker (rabbit, polyclonal IgG, diluted 1:100, Cat# ab15497, Abcam, Cambridge, UK). The formerly mentioned antibodies were applied overnight at 4˚C, then rinsed thoroughly with phosphate-buffered solution to be incubated with the proper biotin-conjugated secondary antibody (Vector Lab. Inc., USA) for sixty minutes at room temperature. The conjugate was visualized by adding chromogen solution (3,3-diaminobenzidine-$H_2O_2$). Slides were counterstained with Mayer's hematoxylin for examination after being washed with distilled water.

**2.7.3. Image acquisition.** The light microscopic sections were examined and captured using a Leica microscope (CH9435 Hee56rbrugg, Switzerland) connected to a digital camera (Leica ICC50 W, 5.0 megapixels). The objective lenses used were plan achromat with the following powers (×10 with scale = 200 μm and NA = 0.25, ×40 with scale = 50 μm and NA = 0.65, and oil lens ×100 with scale = 20 μm and NA = 1.25). The scale bar for each magnification power was inserted into the images. The obtained images were entirely modified concerning color balance, brightness, and contrast without selecting a particular field using the built-in system of MacBook Pro (macOS version 13.4). The authors maintained the original copies.

## 2.8. Morphometric study & image analysis

Per each animal, the most representative ten non-intersecting fields were chosen from each section (magnification power × 400) for the morphometric measurement of the epithelial height in H&E-stained sections, deposited collagen area% in Masson's trichrome-stained sections, and the percentage area of caspase 3, TNF-α, and PCNA reactions in immunohistochemically stained sections utilizing the software image analyzer (Leica Qwin 500, UK).

## 2.9. Transmission Electron Microscope (TEM)

Specimens from the ventral prostate were washed with normal saline, cut into tiny cubes 1 mm$^3$ on ice, and fixed in a cocktail of 2.5% glutaraldehyde, 2.5% paraformaldehyde, and 0.1 M phosphate-buffered solution (pH 7.4). After repeated washing with 0.1 M PBS, the tissues were post-fixed in osmium tetroxide (1%) for 2h at 4˚C. After being dehydrated and cleared, the specimens were embedded in epoxy resin. Semithin sections were cut at 1 μm thickness, stained with toluidine blue, and carefully examined to select specific areas for preparing ultrathin sections. After cutting, the ultrathin sections (70 nm) were harvested on copper grids to be stained with uranyl acetate and lead citrate. Sections were examined under the transmission

electron microscope (JEOL JEM-1200EX 11, Tokyo, Japan) at the Faculty of Science, Ain Shams University, Cairo, Egypt.

### 2.10. Statistical analysis

The biochemical tests' and histomorphometry outcomes proved to be normally distributed and parametric using the Kolmogorov–Smirnov test. Data were abridged as mean and standard deviation. GraphPad Prism software, version 5 (Inc., San Diego, USA), was operated for statistical analysis. The intergroup comparison was performed via One-way ANOVA and Tukey post hoc test, considering the significance at p<0.0001.

## 3. Results

The living conditions in the animal house were generally hygienic, and all medications used were well-tolerated, yielding no deaths among animals in this study.

### 3.1. Effect of DIA on total body weight, prostate weight, and prostate index in testosterone-induced BPH

As shown in Fig 2, the rats' total body weights relatively increased in the BPH group (mean = 235.7±10.35 g) in comparison to the control (mean = 215.4±9.778 g) and the DIA groups (mean = 213.4±5.575 g). At the same time, the total body weight declined in the BPH +DIA group (mean = 219.4±11.83 g) after DIA administration versus the BPH group (Fig 2a). Of note, at p<0.0001, the prostate weight and index showed a substantial increase in the BPH group versus the control and the DIA groups. Whereas the DIA-treated rats in the BPH+DIA group showed considerably regressed prostate weight and prostate index compared with the BPH group (Fig 2b and 2c, respectively).

### 3.2. Effect of DIA on serum testosterone, dihydrotestosterone (DHT), and PSA in testosterone-induced BPH

As depicted in Fig 3, the serum testosterone, DHT, and PSA levels showed a substantial uprise at p<0.0001 in the BPH group versus the control and the DIA groups. The diacerein treatment

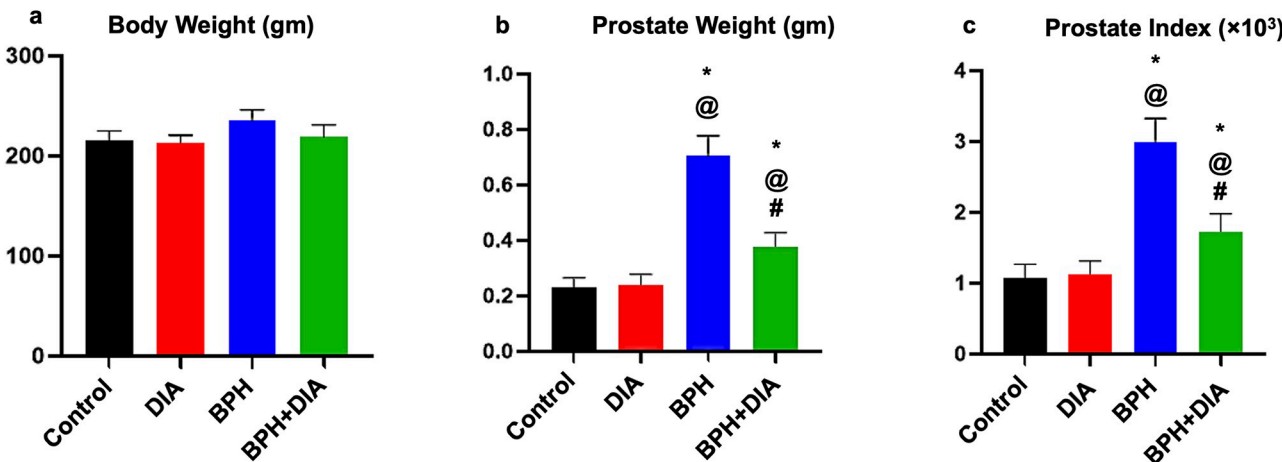

**Fig 2. Statistical analysis of the effect of DIA on (a) rats' body weight, (b) prostate weight, and (c) prostate index in the different research groups.** Data presented as mean ± SD (n = 10). One-way ANOVA and post hoc Tukey test. * significant to the control group, @ significant to the DIA group, # significant to the BPH group, p<0.0001. DIA: diacerein; BPH: benign prostatic hyperplasia.

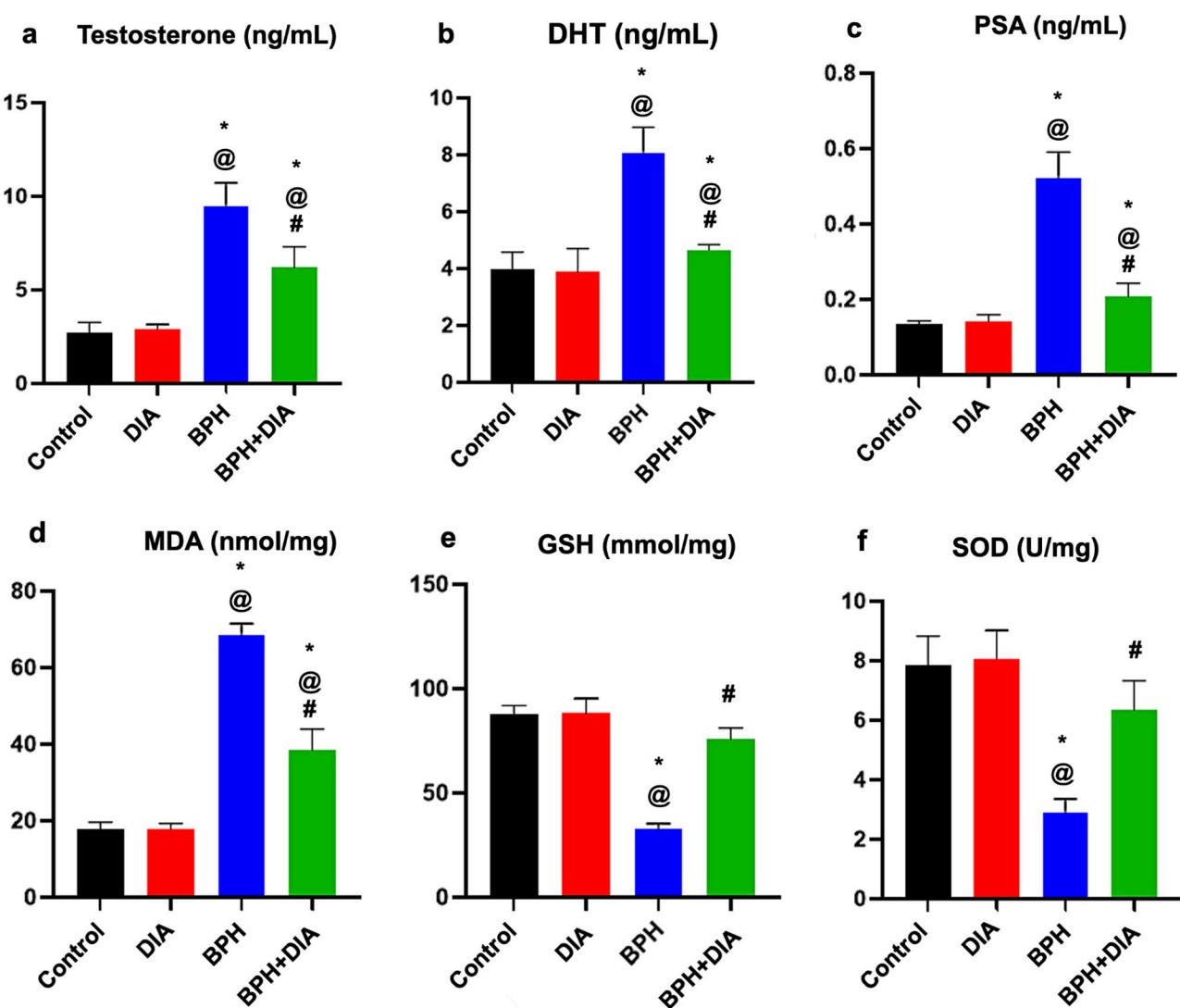

**Fig 3. Statistical analysis of the effect of DIA on serum markers and tissue redox balance.** (a) serum testosterone, (b) serum dihydrotestosterone (DHT), (c) serum PSA, (d) MDA enzyme content, (e) reduced GSH enzyme content, and (f) SOD enzyme content. Data presented as mean ± SD (n = 10). One-way ANOVA and post hoc Tukey test. * significant to the control group, @ significant to the DIA group, # significant to the BPH group, p<0.0001. DIA: diacerein; BPH: benign prostatic hyperplasia.

after induction of BPH has significantly reduced the level of all the previously mentioned serum markers in the BPH+DIA group at p <0.0001 opposite the BPH group.

### 3.3. DIA restored the normal redox balance in prostatic tissue in rats with testosterone-induced BPH

Fig 3 clarifies that at p<0.0001, the induction of BPH resulted in substantially increased oxidant MDA enzyme and significantly regressed levels of antioxidant GSH and SOD enzymes in prostatic tissue homogenate versus the control and the DIA groups, respectively. DIA treatment after BPH induction significantly decreased MDA and significantly increased GSH and SOD levels in the BPH+DIA group at p<0.0001 compared to the BPH group. These outcomes

underscore the putative antioxidant impact of DIA, which restored the average redox balance in prostatic tissue in cases of BPH.

### 3.4. Effect of DIA treatment on the histologic examination of prostatic tissue in testosterone-induced BPH

Microscopic examination of H&E and toluidine blue stained sections from the prostate of the control (Fig 4a–4c) and the DIA groups (Fig 4d–4f) showed variable-sized crammed regular

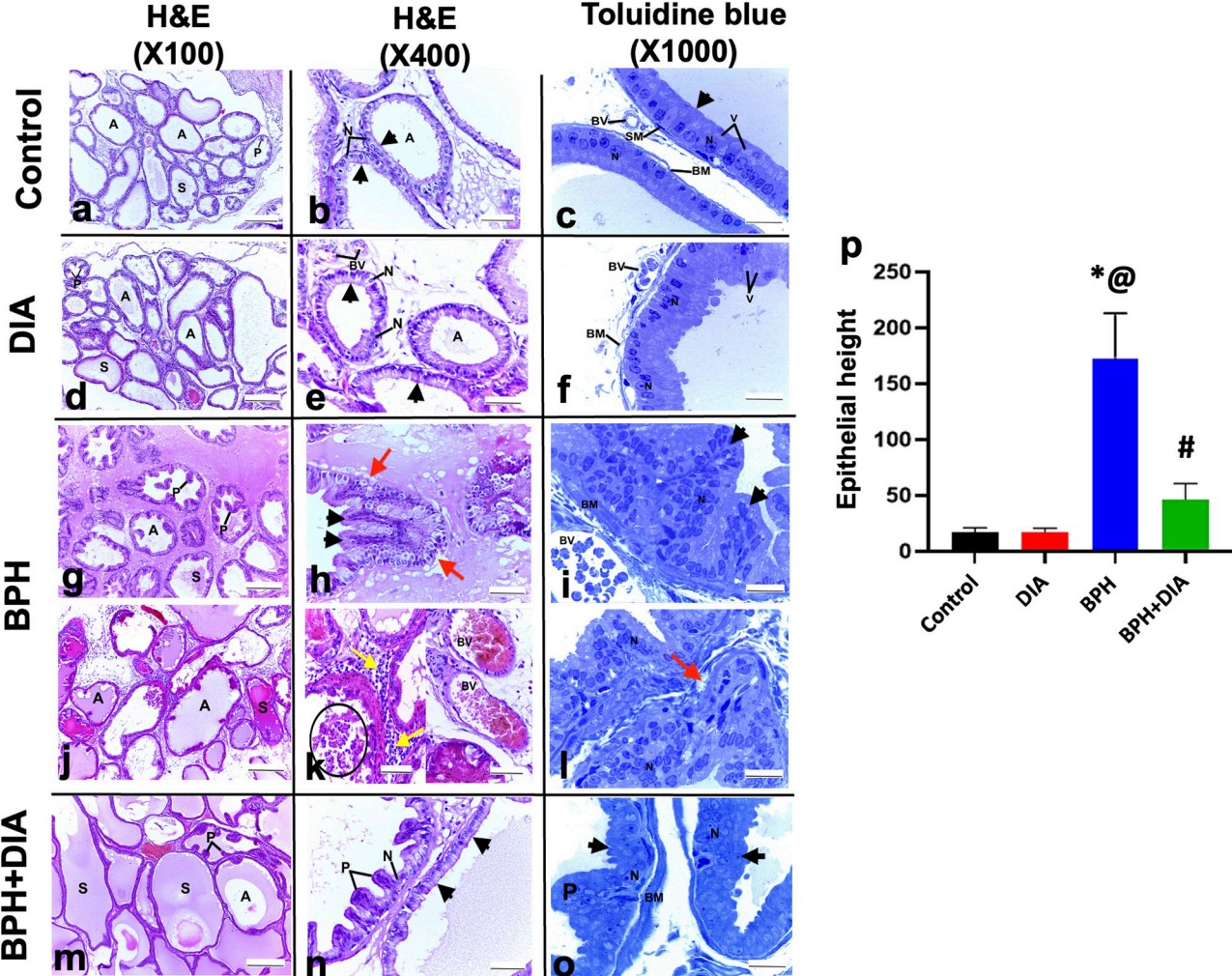

**Fig 4. Photomicrographs of prostate sections stained with H&E and toluidine blue.** (a-c) The control group and (d-f) DIA groups showing normal histology of the prostate with variable-sized crammed regular acini (A) lined by one layer of cuboidal to columnar epithelium (arrowheads) with apical cytoplasmic vacuolation (V) and with basal rounded vesicular nuclei and distinct nucleoli (N). The epithelium is underlined by a clear basement membrane (B.M) and smooth muscle cells (SM). The acinar lumen shows few short papillary projections (P) and eosinophilic secretions (S). The intervening stroma is thin and fibromuscular, containing blood vessels (BV). (g-l) BPH group showing spaced-out pleomorphic acini (A), lined with hyperplastic epithelium (arrowheads) with frequent protruding papillary projections (P). The acinar lumen contains accumulated eosinophilic secretions (S) and sloughed epithelium (circle). The basement membrane is markedly thickened (BM) with focal disruption at some points (red arrows). The intervening stroma is thickened and edematous with markedly congested vasculature (BV) and inflammatory infiltrates (yellow arrows). (m-o) BPH+DIA group with improved prostatic histology showing packed regular acini (A) lined with one or few layers of cuboidal epithelium (arrowheads) resting on intact thickened basement membrane (BM). The acinar epithelial cells show vesicular nuclei with distinct nucleoli (N). The acinar lumen shows a few short projecting papillae (P) and eosinophilic secretions (S). (p) Statistical analysis of the acinar epithelial height in all study groups. Data presented as mean ± SD (n = 10). One-way ANOVA and post hoc Tukey test. * significant to the control group, @ significant to the DIA group, # significant to the BPH group, p<0.0001. DIA: diacerein; BPH: benign prostatic hyperplasia.

prostatic acini with intervening thin fibromuscular stroma containing blood vessels. Acini were lined by a cuboidal to columnar epithelial layer with apically vacuolated cytoplasm and basal rounded vesicular nuclei with distinct nucleoli. The acinar lumen contained eosinophilic secretions and sporadic short papillary projections covered with pseudostratified columnar epithelium. The acinar cells were underlined by smooth muscle cells with a clear basement membrane underneath. The stroma contained blood vessels.

The administration of testosterone for the induction of BPH yielded marked histopathologic changes in the BPH group, which showed spaced-out pleomorphic acini lined by hyperplastic proliferating epithelium with vesicular pleomorphic nuclei forming papillary projections in the acinar lumen. Accumulated eosinophilic secretions and sloughed epithelium were seen in the acinar lumen. The basement membrane was noticeably thickened; however, it was disrupted at some points. The intervening fibromuscular stroma was thickened and edematous with congested vasculature and marked inflammatory infiltrates (Fig 4g–4l).

Notably, the subsequent DIA treatment after induction of PBH imparted a tremendous impact on the BPH+DIA group and markedly alleviated the histological alterations compared to the BPH group. Histologic examination of the BPH+DIA group showed more regular packed acini, mainly lined with single or few layers of cuboidal epithelium with vesicular nuclei and distinct nucleoli resting on an intact, thick basal lamina. The acinar lumen contained a few short projecting papillae and some eosinophilic secretions (Fig 4m–4o). Statistical analysis of the morphometric measurements of the acinar epithelial height showed a substantial rise at $p<0.0001$ in the BPH group compared to the control and DIA groups. In contrast, the epithelial height significantly regressed in the BPH+DIA group versus the BPH group, with no significant difference compared to the control and the DIA groups (Fig 4p).

## 3.5. DIA reduced collagen deposition in prostatic tissue in testosterone-induced BPH

Masson's trichrome-stained sections displayed delicate stromal collagen fibers in the control and DIA groups (Fig 5a and 5b). Many thick, wavy collagen bundles in the BPH group were spotted among the acini and surrounding the blood vessels, significantly higher than the BPH+DIA group at $p<0.0001$ (Fig 5c and 5d). In comparison, DIA treatment succeeded in reducing the collagen fibers in the stroma and around blood vessels in the BPH+DIA group (Fig 5e) with a significant regression compared to the BPH group at $p<0.0001$.

## 3.6. Immunohistochemical examination

**3.6.1. DIA stimulates apoptosis in testosterone-induced BPH via upregulation of caspase 3 expression.** Examination of prostatic immunostained sections with anti-caspase 3 antibodies showed a negative cytoplasmic reaction in the control group (Fig 6a) and the DIA group (Fig 6d). The induction of BPH in rats resulted in mild cytoplasmic expression of caspase 3 without a substantial variance to the control and DIA groups at $p<0.0001$ (Fig 6g). Distinctively, the subsequent DIA treatment in the BPH+DIA group notably accentuated the cytoplasmic caspase 3 expression (Fig 6j) that showed a considerable rise at $p<0.0001$ opposite the control, DIA, and BPH groups.

**3.6.2. DIA attenuates inflammation in testosterone-induced BPH via downregulation of TNF-α expression.** Examination of prostatic immunostained sections with anti-TNF-α antibodies showed negative cytoplasmic TNF-α reaction in the control group (Fig 6b) and the DIA group (Fig 6e). A strong positive cytoplasmic expression of TNF-α was

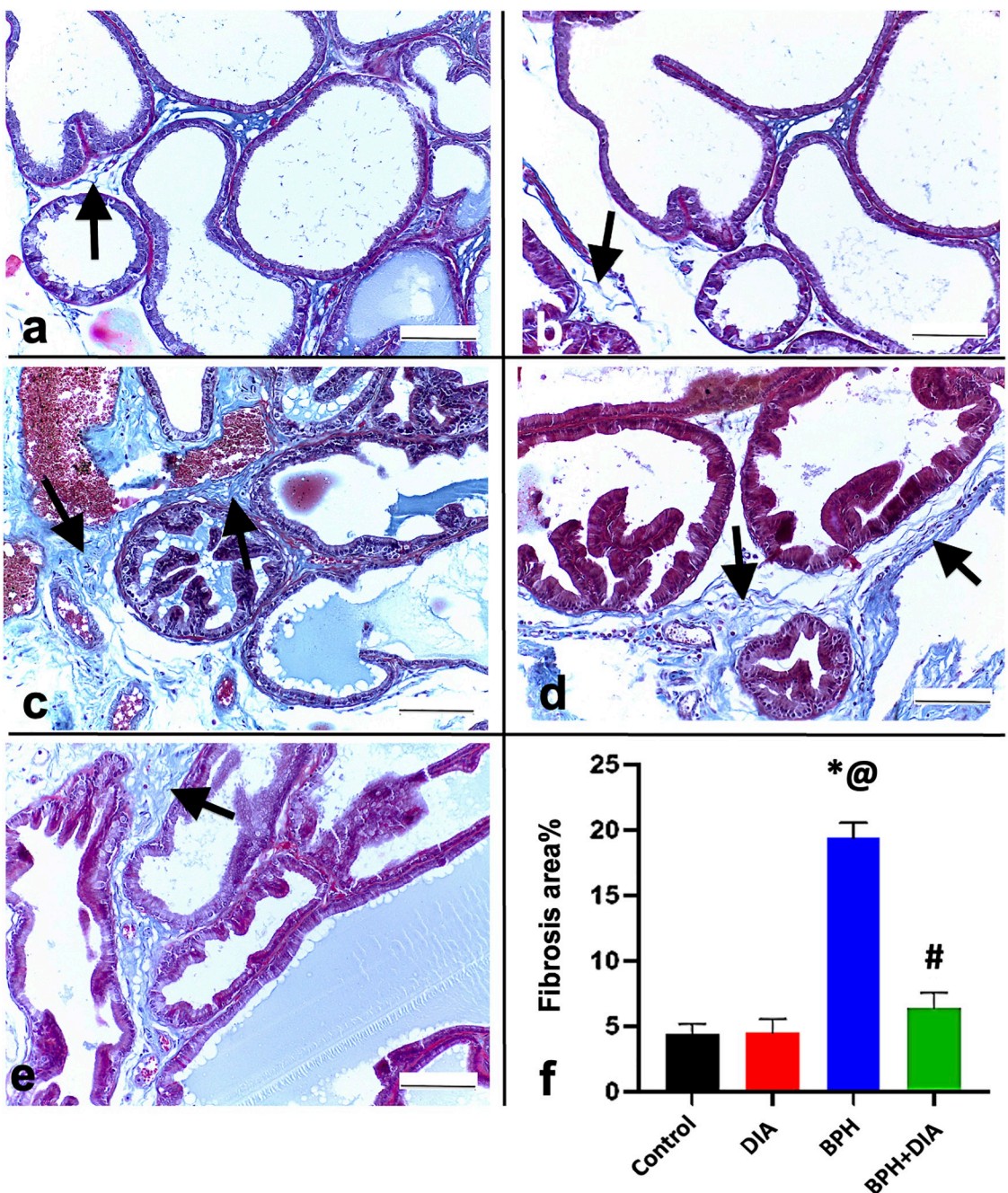

**Fig 5. Photomicrographs of prostate sections stained with Masson's trichrome (Magnification × 400).** (a, b) The control and DIA groups showing delicate collagen fibers in the stroma (arrows). (c, d) BPH group showing thick wavy collagen bundles between prostatic acini and around blood vessels (arrows). (e) BPH+DIA group displaying reduced deposited collagen (arrow). (f) Statistical analysis of the mean area % of deposited collagen in all study groups. Data presented as mean ± SD (n = 10). One-way ANOVA and post hoc Tukey test. * significant to the control group, @ significant to the DIA group, # significant to the BPH group, p<0.0001. DIA: diacerein; BPH: benign prostatic hyperplasia.

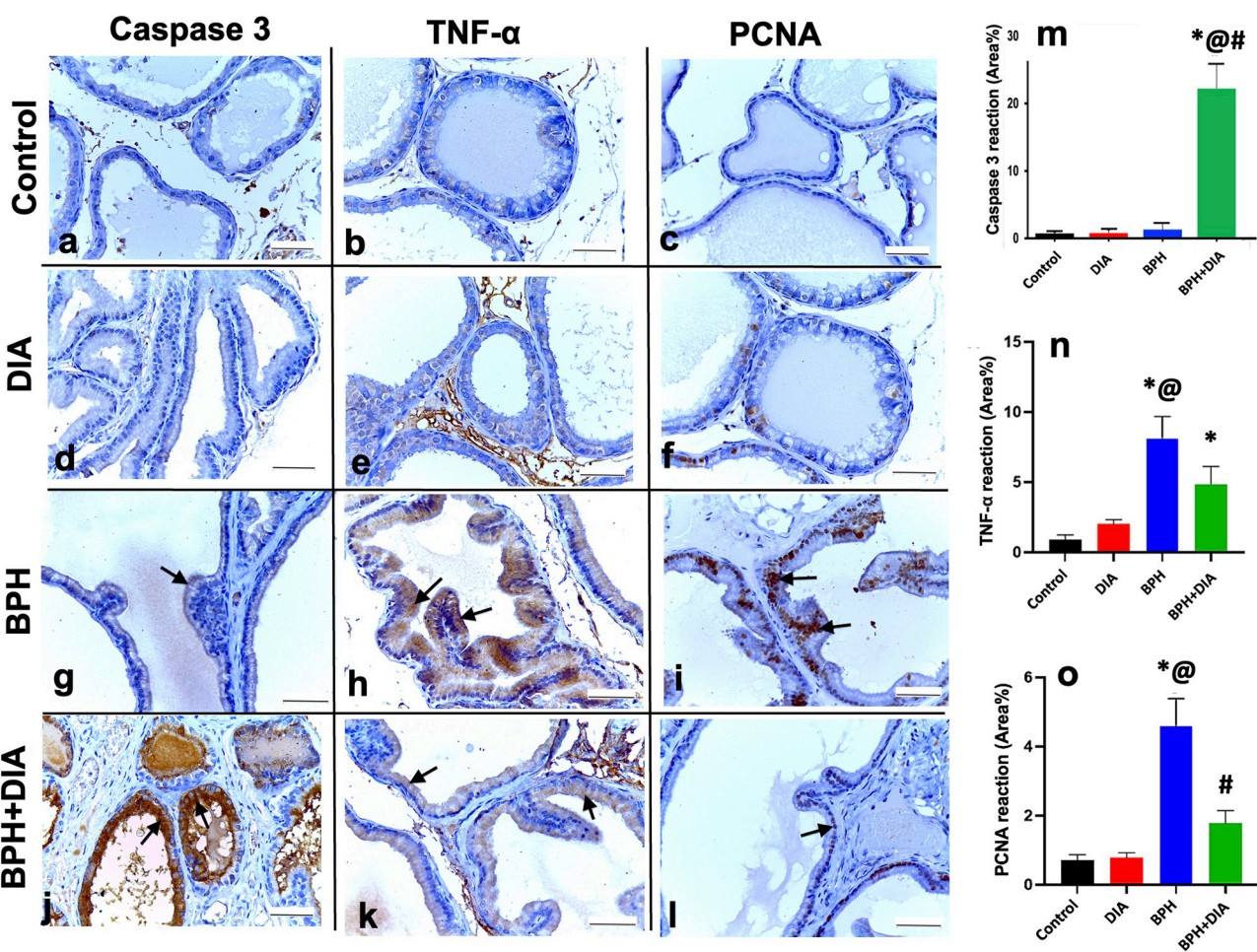

**Fig 6. Photomicrographs of prostate sections immunostained with anti-caspase 3, anti-TNF-α, and anti-PCNA antibodies (Magnification × 400).** (a-c) the control group and (d-f) DIA group showing negative cytoplasmic caspase 3 and TNF-α reactions and minimal nuclear PCNA expression, respectively. (g-i) BPH group showing mild cytoplasmic caspase 3, strong TNF-α reactions, and intense nuclear PCNA expression, respectively (arrows). (j-l) BPH+DIA group exhibiting accentuated cytoplasmic caspase 3, attenuated TNF-α, and mild PCNA expressions, respectively (arrows). (m-o) Statistical analysis of the mean area% of caspase 3, TNF-α, and PCNA immunoreaction in all study groups, respectively. Data presented as mean ± SD (n = 10). One-way ANOVA and post hoc Tukey test. * significant to the control group, @ significant to the DIA group, # significant to the BPH group, p<0.0001. DIA: diacerein; BPH: benign prostatic hyperplasia.

encountered after induction of BPH (Fig 6h) with a substantial increase at p<0.0001 versus the control and DIA groups. In contrast, DIA treatment attenuated the TNF-α expression in BPH+DIA group (Fig 6k) without exhibiting a substantial reduction at p<0.0001 versus the BPH group.

**3.6.3. DIA inhibits epithelial proliferation in testosterone-induced BPH via downregulation of PCNA expression.** Examination of prostatic immunostained sections with anti-PCNA antibodies showed minimal PCNA nuclear reaction in the control group (Fig 6c) and the DIA group (Fig 6f). The BPH group exhibited an intense nuclear PCNA expression, significantly higher than the control and DIA groups at p<0.0001 (Fig 6i). On the other hand, DIA administration in rats suffering from BPH resulted in regressed PCNA expression (Fig 6l), which was considerably lower than the PBH group (p<0.0001) without a tangible difference between the control and the DIA groups.

### 3.7. Effect of DIA treatment on the ultrastructure of prostatic tissue in testosterone-induced BPH

Ultrathin prostatic tissue sections from the control and DIA groups showed acinar cells with basal euchromatic nuclei with heterochromatin islands and regular nuclear membrane. The cytoplasm contained rough endoplasmic reticulum and apical vacuoles. Apical microvilli were seen protruding in the lumen. The acinar cells were resting on a regular basal lamina with smooth muscle cells in between. Delicate collagen fibers were spotted underneath the acinar cells and in the stroma (Fig 7a–7d). Ultrathin sections from the BPH group displayed multiple layers of proliferating acinar cells. The nuclei were euchromatic, with many mitotic figures and irregular nuclear membranes. The basement membrane was markedly thickened with focal disruption. The stroma showed thick bundles of collagen arranged in various directions with hypertrophic active fibroblasts (Fig 7e–7g). Administration of DIA in rats with BPH markedly ameliorated the ultrastructural abnormalities and restored the normal cytoarchitectural pattern of the prostatic tissue. Sections from the BPH+DIA group showed that most prostatic acini were lined by one layer of acinar epithelial cells with normally looking nuclei. The apical parts showed many secretory granules and heterogeneous vacuoles. The apical surface showed projecting microvilli. Notably, the basement membrane was regular and intact, with little collagen fibers spotted above and underneath (Fig 7h–7j).

## 4. Discussion

In the present study, testosterone-induced BPH was confirmed grossly by a significant rise in prostatic weight and index, biochemically by increasing serum levels of testosterone, DHT, and PSA, morphologically by marked acinar epithelial and stromal hyperplasia, inflammatory infiltrates, and increased collagen deposition, together with subsequent rise in the inflammatory TNF-α and the proliferative PCNA markers. Treatment with DIA markedly decreased the prostate weight index and plasma hormones. Moreover, DIA could improve the prostatic tissue histological structure as well as the redox balance. Additionally, DIA could increase the proapoptotic caspase 3 expression besides substantially reducing TNF-α and PCNA expression.

TP was used in the current study to build the rat model with BPH. Androgens were previously shown to have a pivotal role in the etiology of BPH when administered to experimental animals [28]. The prostatic ventral lobe was chosen in our study as it simulated that of the human prostate. Moreover, it was known to be the most androgen-dependent prostatic lobe [29].

Although many therapeutic agents were used to treat BPH, such as 5α-red 2 inhibitors and α-blockers, there was a worldwide concern to introduce new management strategies with fewer side effects [16]. DIA, an anthraquinone derivative, was used in our study owing to its known antioxidant, anti-inflammatory, and antitumor actions [20].

In the present study, BPH was confirmed grossly by the significant rise in the prostatic weight and prostatic index in the BPH group opposite the control group. Histologically, there were spaced-out pleomorphic acini lined by hyperplastic epithelium, whereas their lumina showed protruding papillary projections and accumulated eosinophilic secretions. The intervening fibromuscular stroma was thickened and edematous with congested vasculature and marked inflammatory infiltrates. Former studies supported all these findings and could be explained by cytoplasmic hormonal accumulation [30, 31]. Increased prostatic weight, prostatic index, and epithelial hyperplasia are considered tangible signs in the progression of BPH [32, 33]. These findings could be explained by increased testosterone conversion to DHT due to enhanced 5α-reductase enzyme activity in the hyperplastic prostatic cells [9]. DHT binding

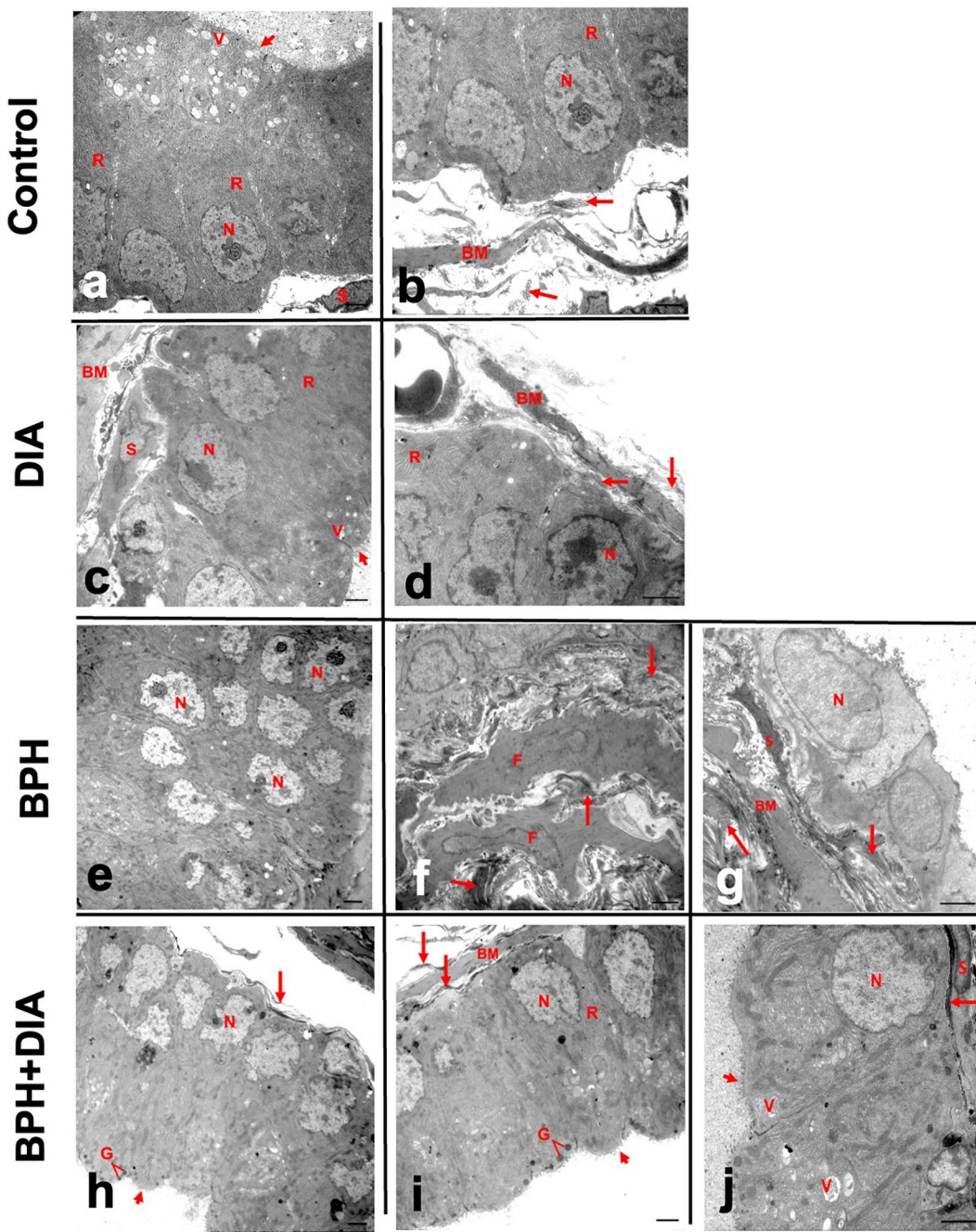

**Fig 7. TEM photomicrographs of ultrathin sections of prostate.** (a, b) control group and (c, d) DIA group showing acinar cells with basal euchromatic nuclei with islands of heterochromatin and regular nuclear membrane resting on regular basement membrane with smooth muscle cells in between. The cytoplasm contains rough endoplasmic reticulum, apical vacuoles, and secretory granules. Apical microvilli are seen protruding in the lumen. Delicate collagen fibers are spotted underneath the acinar cells and in the stroma. (e-g) BPH group showing multiple layers of acinar cells. Nuclei are euchromatic, with many mitotic figures and irregular nuclear membranes. The basement membrane is markedly thickened with focal disruption. The stroma shows thick collagen bundles arranged in various directions with hypertrophic active fibroblasts. (h-j) BPH+DIA group showing a single layer of acinar epithelial cells with normally looking nuclei. The apical parts show many secretory granules and heterogeneous vacuoles. The apical surface shows projecting microvilli. The basement membrane is regular and intact. Little collagen fibers are spotted above and underneath the basement membrane. N: nucleus, R: rER, V: vacuoles, S: smooth muscle, BM: basement membrane, F: fibroblasts, G: secretory granules, long arrow: collagen fibers, short arrow: microvilli. Scale bar: 2 μm.

to nuclear androgen receptors may signal the production of mitogenic growth factors in epithelial and stromal cells [34]. In the present study, we found a significant increase in the serum level of DHT in the BPH group versus the control group. Furthermore, the mean area percent of the stromal deposited collagen increased considerably in the BPH group compared to the control group. Thickening of fibromuscular stroma was also reported in previous models and might be attributed to high serum levels of DHT [35, 36].

The present study used DIA to investigate its therapeutic effect on the rat model of BPH induced by testosterone injection. Our biochemical investigations proved that DIA has dramatically lessened many parameters, such as the level of testosterone, DHT, and PSA in serum, besides reducing prostate weight and epithelial height in rats treated with DIA. PSA is increased in the blood serum due to prostatic trauma as well as the presence of prostatic diseases, for example, prostatitis, BPH, and cancer. At the same time, prostatic epithelial disruption leads to the leakage of PSA into the blood [37].

Many studies confirmed that the progress of prostatic diseases, including BPH, is associated with an alteration in androgen levels. Moreover, the therapeutic influence of any drug used in BPH treatment is highly correlated with its ability to normalize androgen levels [38, 39].

In the present study, DIA treatment markedly alleviated the cytoarchitectural alterations induced by BPH in the prostatic tissues. Also, it was able to lessen the deposited collagen significantly. The tremendous tissue amendment upon using DIA might be attributed to the attenuation of cell proliferation, inflammation, apoptosis, and oxidative stress during the development of BPH [40]. Former studies reported the pivotal role of deranged redox balance accompanying the pathogenesis of BPH. Preserving redox balance is essential for controlling standard cellular functions involving cell proliferation, survival, and aging. Unrestrained ROS formation can eventually result in undue cell proliferation and hyperplasia [41].

TP administration used to build the BPH model in rats was previously proven to generate excessive production of ROS together with attenuation of cellular antioxidant mechanisms [14, 42, 43]. This could justify the increase in oxidant MDA enzyme and the significantly regressed levels of antioxidant GSH and SOD enzymes in prostatic tissue versus the control group in the current study. Also, these findings were per previous studies that had affirmed similar results [44, 45].

In the present work, the subsequent usage of DIA after induction of BPH significantly decreased MDA and significantly increased GSH and SOD levels in the BPH+DIA group compared to the BPH group. These outcomes underscore the impact of DIA in restoring the average redox balance in prostatic tissue in cases of BPH. Previous studies substantiated these results by confirming the antioxidant-modulating effect of DIA in different pathologies [20, 46].

BPH management strategies aimed at reducing inflammation and cellular proliferation, along with induction of apoptosis [47]. It has been reported that TP administration led to an imbalance between the rate of cellular proliferation and apoptosis, whereas proliferation takes the upper hand, resulting in BPH [48]. PCNA, a key factor in mastering DNA duplication and the cell cycle, is linked to prostatic cell proliferation in such a way that prostatic cell cycle arrest is considered a consequence of its absence [49]. In the up-to-date study, the BPH group exhibited mild cytoplasmic caspase 3 and intense nuclear expression of PCNA. On the other hand, the subsequent DIA treatment notably accentuated the cytoplasmic caspase 3 expression together with a marked decline in the nuclear PCNA expression. These findings aligned with Bharti and colleagues, who found that DIA could activate caspase-dependent apoptosis in breast cancer [50]. Another explanation for the disparity between the scale of proliferation and apoptosis might be attributed to the enhanced telomerase activity within the stem cells of prostatic stroma in the BPH model [51]. Telomerase plays a chief role in the maintenance of

telomere length by preventing its shortening during the cell cycle. At the same time, telomerase activation might promote proliferation more than apoptosis [52].

Furthermore, other studies assumed a crucial link between cellular proliferation in BPH and chronic prostatic inflammation, which in turn led to a subsequent accumulation of inflammatory cells with increased expression of cytokines. These cytokines might contribute to the development and progression of BPH [12, 53]. Such inflammation in BPH may disrupt the cellular connections, leading to abnormalities in the basement membrane and allowing secretory proteins to permeate from the epithelium to the stroma [54]. This might be the cause of the present study's marked thickening and focal disruption of the basement membrane. In agreement, we also reported an upregulation of TNF-α in the BPH group versus the control group in our study. Whereas DIA treatment, via its anti-inflammatory effects, could successfully lessen TP-induced upregulation of TNF-α. This data agreed with former studies suggesting the anti-inflammatory action of DIA [13, 55].

## 5. Conclusion

Our study highlighted the DIA-induced remarkable improvement in the histopathological characteristics of prostatic tissue, in addition to an optimistic regression of prostate enlargement in the rat model of BPH. According to a certain hypothesis, the potent antioxidant, antiproliferative, anti-inflammatory, and apoptotic-inducing effects of DIA cause its antihyperplastic effects via upregulation of caspase 3 and downregulation of TNF-α and PCNA as well as restoring the normal redox and hormonal balance. Undoubtedly, in-depth molecular studies are still needed to further explore the exact pathophysiological mechanisms and pathways of DIA action, the point which may be challenging to many researchers.

## Supporting information

**S1 Graphical abstract.**
(TIFF)

## Author Contributions

**Conceptualization:** Azza Saleh Embaby, Mai A. M. Almoatasem.

**Data curation:** Rabab Ahmed Rasheed, Dina S. Hussein.

**Funding acquisition:** Mohamed M. A. Elshaer, Saeedah Musaed Almutairi.

**Investigation:** Rabab Ahmed Rasheed, A. S. Sadek, R. T. Khattab, Fatma Alzahraa A. Elkhamisy, Dina S. Hussein, Azza Saleh Embaby.

**Methodology:** Rabab Ahmed Rasheed, A. S. Sadek, Fatma Alzahraa A. Elkhamisy, Heba Abdelrazak Abdelfattah, Azza Saleh Embaby, Mai A. M. Almoatasem.

**Resources:** Saeedah Musaed Almutairi, Mai A. M. Almoatasem.

**Software:** Rabab Ahmed Rasheed, Heba Abdelrazak Abdelfattah.

**Supervision:** Rabab Ahmed Rasheed, R. T. Khattab, Mohamed M. A. Elshaer.

**Validation:** Rabab Ahmed Rasheed.

**Visualization:** Rabab Ahmed Rasheed, Fatma Alzahraa A. Elkhamisy, Dina S. Hussein.

**Writing – original draft:** A. S. Sadek, R. T. Khattab, Heba Abdelrazak Abdelfattah, Mohamed M. A. Elshaer, Mai A. M. Almoatasem.

Writing – review & editing: Rabab Ahmed Rasheed, Saeedah Musaed Almutairi.

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
