## [Decision Letter · Decision Letter 0]

11 Sep 2023

PONE-D-23-23328Diacerein provokes apoptosis, improves redox balance, and downregulates PCNA and TNF-α in a rat model of testosterone-induced benign prostatic hyperplasia: a new non-invasive approachPLOS ONE

Dear Dr. Rasheed,

Thank you for submitting your manuscript to PLOS ONE. After careful consideration, we feel that it has merit but does not fully meet PLOS ONE’s publication criteria as it currently stands. Therefore, we invite you to submit a revised version of the manuscript that addresses the points raised during the review process.

We look forward to receiving your revised manuscript.

Kind regards,

Abeer El Wakil, PhD

Academic Editor

PLOS ONE

Journal Requirements:

"The authors extend their appreciation to the Researchers supporting project number (RSP2023R470) King Saud University, Riyadh, Saudi Arabia."

"The authors received no specific funding for this work"

**Additional Editor Comments:**

Dear Authors,

The concept of the present study seems interesting, but the whole manuscript doesn’t have enough quality for publication in Plos One journal. However, I believe that the work itself is relevant and I have therefore provided feedback on how to improve the manuscript to be acceptable for publication. Unfortunately, the manuscript has fundamental and major issues. They are mentioned below:

- The abstract is informative and completely self-explanatory. However, I believe that it would be better to be concise.

- Keywords: Keywords enable authors to extend the representation of their manuscript content beyond that presented in the title and abstract. Keywords make your paper searchable and ensure that you get more citations. Thus, it is important to include the most relevant keywords that will help other authors find your paper. It is therefore very important to avoid repeating keywords already cited in the title to increase the discoverability of your paper. Authors need to revise and address this point.

- Abbreviations must be used only if the word is repeated more than 3 times in your text, otherwise there is no need to use abbreviation. Also, the abbreviation must be cited in the first position where the full word exists, then only the abbreviation is written afterwards according to applicable international standards and rules. Authors need to address this regarding phosphate- buffered solution (PBS) (check lines 173, 214, and 223).

- The major weak point of this manuscript is the figures' presentation. They are not presented in an insightful way. Some figures could be grouped, for example, Fig.3+Fig.4; Fig. 5+Fig. 6; Fig. 8+ Fig. 9 + Fig.10. For the semi-thin sections, Fig. 6e do not add new information, so the authors could delete it to have symmetrical figure if they are going to group it with Fig.5.

- At line 156, the authors mentioned that animals "left without treatment for another 2 weeks". However, in the graphical abstract, they mentioned that animals received 0.5% CMC-Na solution for 2 weeks. Adjust.

- Although the paper is generally well written and the English language is appropriate and understandable, I recommend editing the main text for English language and grammar to improve readability and clarity for our readers. This can be done by a native English speaker or a professional editing service.

Minor points to be addressed:

- The measurement units should be written according to the International Standards. Consequently, I suggest to replace "ml" with "mL" throughout the whole manuscript.

- In the Materials and Methods section, authors must start with "Drugs and Materials".

- Replace PH at line 173 by pH.

- Replace BPS at line 223 by PBS.

- Replace gm by g for grams (even if some papers use gm, it is not common).

Reviewers' comments:

Reviewer's Responses to Questions

**Comments to the Author**

1. Is the manuscript technically sound, and do the data support the conclusions?

Reviewer #1: Partly

2. Has the statistical analysis been performed appropriately and rigorously? 

Reviewer #1: Yes

3. Have the authors made all data underlying the findings in their manuscript fully available?

Reviewer #1: Yes

4. Is the manuscript presented in an intelligible fashion and written in standard English?

Reviewer #1: No

5. Review Comments to the Author

Reviewer #1: The present manuscript is claimed that Diacerein provokes apoptosis, improves redox balance, and downregulates PCNA and TNF-α in a rat model of testosterone-induced benign prostatic hyperplasia.

1- There are various typo errors in the text.

2- In the title please clarify these change took place in prostate.

3-introduction is too weak. Superficial information was introduced. You should expand introduction in case of role of apoptosis, inflammation as well as pcna in the pathophysiology of bph. Pharmacological propertied of DIA should be explain.

4-What is your rational for dose and time of agent administrations?

5-discussion should be begin with your core finding.

6. PLOS authors have the option to publish the peer review history of their article (what does this mean?). If published, this will include your full peer review and any attached files.

Reviewer #1: No

---

## [Author Response · Author response to Decision Letter 0]

21 Sep 2023

Many thanks to the respectful editor and reviewer for reviewing our manuscript to improve its quality to be valid for publication in your esteemed journal. Kindly find the authors' response to the editor and the reviewer's recommendations.

 Editor Comment Action

1- Please ensure that your manuscript meets PLOS ONE's style requirements, including those for file naming. Done

2- Please remove any funding-related text from the manuscript and let us know how you would like to update your Funding Statement.

Please include your amended statements within your cover letter; we will change the online submission form on your behalf. The funding-related text was removed from the acknowledgment section; the cover letter included the amendments.

3- The abstract is informative and completely self-explanatory. However, I believe that it would be better to be concise. Thank you very much. The abstract was reformulated to be more concise.

4- Avoid repeating keywords already cited in the title to increase the discoverability of your paper Keywords have been updated. 

5- The abbreviation must be cited in the first position where the full word exists. Authors need to address this regarding phosphate-buffered solution (PBS) (check lines 173, 214, and 223). Thank you very much. The abbreviations mentioned have been corrected according to the editor's recommendations. 

6- Some figures could be grouped, for example, Fig.3+Fig.4; Fig. 5+Fig. 6; Fig. 8+ Fig. 9 + Fig.10. Thank you very much. Figures have been merged according to the editor's instructions, and the figures' legends have been modified.

7- For the semi-thin sections, Fig. 6e do not add new information, so the authors could delete it to have symmetrical figure if they are going to group it with Fig.5. Thank you very much. We apologize that we couldn't delete Fig 6e as it is the only photo representing the treated group (DIA+BPH) in the semithin sections. However, we merged Fig 5 + Fig 6 according to the editor's suggestions. 

8- At line 156, the authors mentioned that animals "left without treatment for another 2 weeks". However, in the graphical abstract, they mentioned that animals received 0.5% CMC-Na solution for 2 weeks. Adjust. Thank you for the precious note. The authors adjusted the experimental design in the manuscript. The animals in the BPH group received the vehicle (0.5% CMC-Na solution via oral gavage) for a further two weeks after the induction of BPH to mimic the control group. Still, they didn't receive diacerein as a drug treatment. 

9- I recommend editing the main text for English language and grammar to improve readability and clarity for our readers. This can be done by a native English speaker or a professional editing service. Thank you. We revised and edited the manuscript in terms of English language and grammar.

10- Replace "ml" with "mL" throughout the whole manuscript Done. Thank you very much. 

11- In the Materials and Methods section, authors must start with "Drugs and Materials". Thank you for the remark. We started the Material and Methods section with "Drugs and materials".

12- Replace PH at line 173 by pH. Done. Thank you very much.

13- Replace BPS at line 223 by PBS. According to the editor's instructions, the abbreviation has been removed and replaced by the complete word.

14- Replace gm by g for grams Thank you. gm has been replaced by g throughout the whole manuscript.

Reviewer 1

 Reviewer Comment Action

1- There are various typo errors in the text. Thank you very much. We apologize for this. The manuscript has been revised and edited.

2- In the title please clarify these change took place in prostate. Many thanks for the precious note. We included the name of the disease, "Benign Prostatic Hyperplasia," in the title of the manuscript. In addition, we added the word "prostate" in the keywords. 

3- introduction is too weak. Superficial information was introduced. You should expand introduction in case of role of apoptosis, inflammation as well as pcna in the pathophysiology of bph. Pharmacological propertied of DIA should be explain. Thank you very much for the precious advice. The introduction section has been supported by information concerning the role of apoptosis, inflammation, and PCNA in the pathophysiology of BPH. In addition, we referred to the pharmacological and molecular action of DIA.

4- What is your rational for dose and time of agent administrations? Thank you very much. We provided the answer to this point at the end of the text.

5- discussion should be begin with your core finding. Thank you very much. The discussion was reformulated to start with the core findings of our study.

The rationale for the dose and time of testosterone propionate:

We built up the animal model of BPH by daily subcutaneous injection of testosterone propionate at a dose of 5 mg/kg/day for four weeks, as per many previous studies which have followed the same protocol for the induction of BPH (links given below). In addition, as shown in the figure below, the authors were guided by a pilot study that showed that the prostatic enlargement was evidenced by increased prostate weight fourfold after using the previously assigned dose for four consecutive weeks.

1. https://doi.org/10.1371/journal.pone.0236879

2. https://doi.org/10.1155/2021/2094665

3. https://doi.org/10.1038/s41598-019-56145-z

The rationale for the dose and time of testosterone propionate:

For treating BPH, we used DIA at a dose of 50 mg/kg daily via oral gavage for two weeks. Almezgagi and colleagues1 affirmed that DIA acts as an antioxidant, anti-inflammatory, and antiapoptotic when used for 2-4 weeks at the previously mentioned dose. Similarly, DIA was used in the same dose in rats for 15 days to protect against doxorubicin-induced nephrotoxicity and for 7 days to protect against glycerol-induced acute renal injury via modulating the oxidative stress, apoptotic activities, and inflammatory mediators2,3. In addition, Malaguti and colleagues proved in their study that DIA could downregulate the level of proinflammatory cytokines (IL-1β, TNF-α, IFN-γ, and IL-12) in diabetic mice when injected i.p at the same dose for 24 days. As shown in the previous figure, our pilot study showed that DIA had a noticeable ameliorative effect on the prostate when administered to rats at the previously mentioned dose and duration. 

1. https://doi.org/10.1016/j.biopha.2020.110594

2. https://doi.org/10.1155/2016/9507563

3. https://doi.org/10.1016/j.intimp.2008.01.020

4. https://doi.org/10.1016/j.intimp.2008.01.020

Depending upon the formerly mentioned studies and the results of our pilot study, the authors concluded that the used dose and duration are the most appropriate for the induction and treatment of BPH in rats.

---

## [Decision Letter · Decision Letter 1]

16 Oct 2023

Diacerein provokes apoptosis, improves redox balance, and downregulates PCNA and TNF-α in a rat model of testosterone-induced benign prostatic hyperplasia: a new non-invasive approach

PONE-D-23-23328R1

Dear Dr. Rasheed,

We’re pleased to inform you that your manuscript has been judged scientifically suitable for publication and will be formally accepted for publication once it meets all outstanding technical requirements.

Kind regards,

Abeer El Wakil, PhD

Academic Editor

PLOS ONE

Additional Editor Comments (optional):

The authors have satisfactorily addressed the concerns raised by the editor and the reviewer.

Reviewers' comments:

Reviewer's Responses to Questions

**Comments to the Author**

1. If the authors have adequately addressed your comments raised in a previous round of review and you feel that this manuscript is now acceptable for publication, you may indicate that here to bypass the “Comments to the Author” section, enter your conflict of interest statement in the “Confidential to Editor” section, and submit your "Accept" recommendation.

Reviewer #1: All comments have been addressed

2. Is the manuscript technically sound, and do the data support the conclusions?

Reviewer #1: Yes

3. Has the statistical analysis been performed appropriately and rigorously? 

Reviewer #1: Yes

4. Have the authors made all data underlying the findings in their manuscript fully available?

Reviewer #1: Yes

5. Is the manuscript presented in an intelligible fashion and written in standard English?

Reviewer #1: Yes

6. Review Comments to the Author

Reviewer #1: All comments raised by reviewer have been addressed. Now manuscript is appropriate for publication in the Plos One.

7. PLOS authors have the option to publish the peer review history of their article (what does this mean?). If published, this will include your full peer review and any attached files.

Reviewer #1: No

---

## [Editor Report · Acceptance letter]

31 Oct 2023

PONE-D-23-23328R1 

Diacerein provokes apoptosis, improves redox balance, and downregulates PCNA and TNF-α in a rat model of testosterone-induced benign prostatic hyperplasia: a new non-invasive approach 

Dear Dr. Rasheed:

I'm pleased to inform you that your manuscript has been deemed suitable for publication in PLOS ONE. Congratulations! Your manuscript is now with our production department. 

Kind regards, 

on behalf of

Professor Abeer El Wakil 

Academic Editor

PLOS ONE